# Therapeutic Potential for Targeting Autophagy in ER+ Breast Cancer

**DOI:** 10.3390/cancers14174289

**Published:** 2022-09-01

**Authors:** Ryan M. Finnegan, Ahmed M. Elshazly, Patricia V. Schoenlein, David A. Gewirtz

**Affiliations:** 1Department of Microbiology & Immunology, Virginia Commonwealth University, Richmond, VA 23298, USA; 2Departments of Pharmacology & Toxicology & Medicine, Massey Cancer Center, Virginia Commonwealth University, Richmond, VA 23298, USA; 3Department of Pharmacology and Toxicology, Faculty of Pharmacy, Kafrelsheikh University, Kafrelsheikh 33516, Egypt; 4Department of Cellular Biology and Anatomy, Medical College of Georgia at Augusta University, Augusta, GA 30912, USA

**Keywords:** autophagy, resistance, endocrine, estrogen, breast cancer, cytoprotective

## Abstract

**Simple Summary:**

While ER+ breast cancer is generally considered to have a better prognosis than other breast cancer subtypes, relapse may nevertheless occur years after diagnosis and treatment. Despite initially responding to treatment, 30–40% of tumors acquire resistance to treatment that contributes to disease recurrence, metastasis, and ultimately, death. In the case of the individual estrogen antagonists or aromatase inhibitors, the autophagy induced by these agents is largely cytoprotective. However, whether autophagy inhibition will prove to be a useful strategy for improving outcomes for current combination therapeutic strategies awaits further studies.

**Abstract:**

While endocrine therapy remains the mainstay of treatment for ER-positive, HER2-negative breast cancer, tumor progression and disease recurrence limit the utility of current standards of care. While existing therapies may allow for a prolonged progression-free survival, however, the growth-arrested (essentially dormant) state of residual tumor cells is not permanent and is frequently a precursor to disease relapse. Tumor cells that escape dormancy and regain proliferative capacity also tend to acquire resistance to further therapies. The cellular process of autophagy has been implicated in the adaptation, survival, and reactivation of dormant cells. Autophagy is a cellular stress mechanism induced to maintain cellular homeostasis. Tumor cells often undergo therapy-induced autophagy which, in most contexts, is cytoprotective in function; however, depending on how the autophagy is regulated, it can also be non-protective, cytostatic, or cytotoxic. In this review, we explore the literature on the relationship(s) between endocrine therapies and autophagy. Moreover, we address the different functional roles of autophagy in response to these treatments, exploring the possibility of targeting autophagy as an adjuvant therapeutic modality together with endocrine therapies.

## 1. Introduction

In 2022, approximately 287,850 new cases of invasive breast cancer will occur in women in the US, resulting in 43,250 deaths [1]. Among these, estrogen receptor alpha positive (ER+) breast cancer is the most common disease subtype and is anticipated to constitute approximately 70% of the breast cancer cases [2]. Initial treatment for this breast cancer subtype is endocrine and adjuvant therapy, which has reduced relapse and mortality by almost 40% [3]. There are numerous clinically available endocrine therapies, including selective estrogen receptor modulators (SERMS) such as tamoxifen (TAM), selective estrogen receptor degraders (SERDs), such as Fulvestrant [4] and aromatase inhibitors (AIs). TAM, one of the oldest and most frequently utilized SERMs, is now typically prescribed to treat hormone receptor-positive, early-stage breast cancer after surgery to reduce disease recurrence in pre-menopausal women. Currently, AIs, which show increased efficacy compared to TAM therapy [3] are the preferred endocrine treatment for post-menopausal women with all stages of ER+ breast cancer. Unfortunately, while treatment of early ER-positive breast cancer with SERMs, SERDs, and AIs can reduce recurrence for up to 5 years, resistance to hormone therapy is common, and most cases eventually result in metastatic disease progression [5,6]. While endocrine therapies remain the standard of care for ER+ breast cancer, a major drawback to the success of these therapies is the development of resistance.

## 2. General Mechanisms of Resistance to Endocrine Based Therapies

Resistance is classified as either intrinsic (de novo) or acquired, depending on whether the tumor cells show resistance at the onset of treatment or develop resistance during the therapy. For ER+ breast cancers, intrinsic endocrine resistance is most commonly evident in patients undergoing TAM therapy, with approximately 10% of breast cancers showing initial resistance to TAM. This is frequently due to the patient’s inability to convert TAM to its most active metabolite, endoxifen, due to the presence of inactive alleles of cytochrome P450/2D6 (CYP2D6) [7]. Acquired resistance, on the other hand, regularly occurs in response to SERMs, SERDs, and AIs, and can result from any one of multiple mechanisms that have been elucidated by preclinical and clinical studies [7,8,9,10,11]. Acquired mechanisms of resistance include epigenetic and genetic-based alterations in co-regulators and other transcriptional regulators that target the estrogen receptor alpha (ERα). ERα primarily mediates the proliferative effects of estrogen, whereas ERβ is typically considered antiproliferative in its action [12]. The ERα itself is subject to mutation, particularly in response to AI treatment [13], or to loss over time, which occurs in approximately 20% of breast cancer patients undergoing endocrine treatment [14]. 

ERα genomic and non-genomic actions also can be modified by multiple tyrosine kinase receptor pathways, including HER2, EGFR, FGFR, and IGFIR, that often converge on the activation of the PI3K/AKT and MEK/MAPK1/ 2 survival pathways, as detailed in a recent review [15]. The PI3K catalytic alpha subunit (PIK3CA) itself is commonly mutated in ER+ breast cancer, resulting in AKT hyperactivation [16]. The selective molecular targeting of the receptors and survival signaling pathways implicated in endocrine resistance is being explored as an approach to improve the efficacy of hormonally therapy [17]. However, the targeting of many of these molecular targets, particularly that of growth factor signaling pathways [18] along with endocrine treatments (discussed below) is predicted to induce autophagy, which is otherwise a normal physiological pathway involved in cellular homeostasis. 

Autophagy is a dynamic physiological process that can be induced to protect normal and cancer cells from death (apoptosis) during times of increased cellular stress such as nutrient deprivation, growth factor deprivation, oxidative damage, and hypoxia [19]. In cancer cells, autophagy can provide protection from cell death for a sustained period during which the cells can adapt by genetic or epigenetic mechanisms. Thus, it is not surprising that pre-clinical studies have shown induction of autophagy to facilitate the emergence of antiestrogen-resistant breast cancer cells [20,21]. A protective role for autophagy has also been identified in breast cancer cells undergoing multiple modes of therapy, including growth factor receptor blockade [22], chemotherapy, and radiation therapy [23]. Whether breast cancer cells utilize a common mechanism of autophagy induction in response to the various treatments is not entirely clear. Currently, however, clinical trials are targeting the basic machinery of autophagy (described below) with the goal of improving the response to breast cancer therapies.

## 3. Mechanisms of Autophagy

In mammalian cells, there are three main classifications of autophagy: macroautophagy, microautophagy, and chaperone-mediated autophagy (CMA). Each of these pathways, although morphologically distinct, ultimately involves the delivery of cargo to the lysosome. The lysosome provides the acidic environment and enzymes necessary for the degradation and recycling of cargo [24]. Macroautophagy is the autophagy pathway most commonly implicated in cancer cell resistance to therapy, including hormonal therapy resistance [23]. Macroautophagy is an evolutionarily conserved catabolic process through which cellular cargo is initially sequestered within a double membrane vesicle, prior to fusion with the lysosome (Figure 1). In this regard, macroautophagy is distinct from chaperone-mediated autophagy and microautophagy in that each of these types of autophagy does not rely on an autophagosome to bring cargo to the lysosome [24]. To date, the components of autophagy and the required autophagic machinery are encoded by 31 autophagy-related genes (ATG). Many of these genes and the autophagy pathway itself have been shown to be a necessary component of a number of cellular processes such as immune cell development, maintaining cell and tissue homeostasis, cellular metabolism, aging, and cancer [25]. Thus, it is not surprising that the impairment of autophagy in normal cells has been associated with multiple disease processes [26,27]. 

The autophagy pathway is typically divided into separate stages: initiation of the double membrane phagophore, elongation and closure of the autophagosome membrane, fusion with the lysosome forming the autolysosome, and degradation of the intravesicular cargo. The initiation phase is regulated by the mammalian target of rapamycin, mTOR, which is a central component of two multiprotein complexes, designated mTORC1 and mTORC2. The mTORC1 complex is highly responsive to nutrient deprivation and limited amino acid availability, while mTORC2 responds to growth factor availability. When mTORC1 activity is low, ULK1/2 (Unc-51-like kinase 1/2) is activated via dephosphorylation. The ULK1/2 complex is comprised of ULK1/2, FIP200, and ATG13. This complex, once assembled, phosphorylates members of class III PI3K complex, consisting of AMBRA1, Beclin1, VPS15/34, UVRAG, and ATG14 [28]. Phosphorylation of PI3KC, Beclin-1, and VPS34 is required for the initiation of phagophore nucleation, which is hypothesized to originate from multiple membrane sources, including the endoplasmic reticulum, mitochondria, Golgi apparatus, and recycling endosomes [29,30,31,32]. 

Following the initial nucleation step, the phagophore is elongated by the ATG5/12 complex, which is conjugated by ATG16L and by the conjugation of active cytosolic LC3-I (encoded by ATG8) to phosphatidylethanolamine (PE), generating LC3-II. Conjugation with PE requires sequential activation of ATG7, ATG3, and the ATG5/12 complex [33]. Prior to LC3-I conjugation, cleavage of the C-terminal region of the inactive proform of LC3 is mediated by ATG4B protease. LC3-II is recruited to the phagophore membrane and is required for elongation of the inner and outer membranes of the autophagosome. Following phagophore maturation, the autophagosome fuses with the lysosome, resulting in the formation of an autolysosome, leading to the degradation of the autophagic cargo, along with LC3-II. Thus, LC3-II turnover is often utilized as a marker for autophagosome formation and functional autophagic flux [34]. 

There are five members of the LC3 gene family, with LC3B being the most commonly studied endogenous autophagic marker [35]. LC3-II can be generated from LC3B, LC3A, and LC3C, and antibody specificity is often not confirmed in studies. Whether LC3A-II and LC3B-II differentially impact autophagy function in a cell and context-dependent manner is not fully understood, however, LC3A, LC3B, and LC3C have been identified as having distinct subcellular distributions, kinetics, and expression in cancer cells [35]. In addition to LC3-II, the protein Sequestosome 1 (p62/SQSTM1), a ubiquitin and LC3 binding protein, is also degraded during autolysosomal turnover. Thus, p62 levels and flux can provide an independent measure of functional autophagy. Although p62/SQSTM1 plays a key role in clearing protein aggregates in cells (termed aggrephagy), it is also a selective autophagy receptor and facilitates mitochondrial and lipid droplet turnover specifically termed mitophagy and lipophagy, respectively. Additionally, p62 has roles in the ubiquitin-proteasome system, cellular metabolism, signaling, and apoptosis [36]. 

Although the precise differences between basal and stimulus-induced autophagy are still being clarified, a key point of regulation involves the mTORC1 and mTORC2 complexes. These complexes are nutrient/energy/redox sensors and ultimately regulate protein translation and energy supply in cells. The mTORC1 is particularly sensitive to nutrient deprivation with the regulation of protein synthesis as its main function via 4E-BP1 and S6K. The mTORC2 is a sensor of growth factors, responsive to PI3K, and involved in cellular metabolism. However, there is an overlap in the ability of these complexes to respond to cellular stressors. Also, mTORC1 and mTORC2 can regulate each other. For example, the phosphorylation of PRAS40, a component of mTORC1, is regulated by AKT while Sin1, a component of mTORC2 is regulated by S6K. In part, the regulation of these complexes relies on the two binding proteins RAPTOR and RICTOR. RAPTOR (Rapamycin-sensitive adapter protein of mTOR), promotes the formation of mTORC1, is required for mTORC1 kinase activity, and can determine the subcellular localization of mTORC1. RICTOR (rapamycin-insensitive companion for mTOR) is required for substrate recruitment in forming mTORC2. The binding of RICTOR and RAPTOR to mTOR is mutually exclusive. 

The molecular details of the regulation of mTORC1 and mTORC2 complexes have been recently reviewed [37,38]. Overall, the current consensus is that mTORC1 and mTORC2 may signal in parallel in many cellular contexts. However, there are stresses that selectively activate or repress mTORC1 and mTORC2. For example, energy depletion that upregulates the AMP-activated protein kinases (AMPK) upregulates mTORC2 via phosphorylation of RICTOR [39]. Activation of mTORC2 can also occur indirectly as a consequence of AMP-mediated inactivation of mTORC1 via inhibitory phosphorylation of RAPTOR [40]. AMPK can also directly phosphorylate ULK1, one of the 31 autophagy genes discussed above, and AMPK-mediated phosphorylation appears to regulate the localization of components of the phagophore [41]. The canonical pathway of AMPK activation occurs in response to an energy deficit. AMPK is bound by AMP and ADP, becomes allosterically modified, and is activated by phosphorylation via multiple kinases, including LKB1 [42]. However, recent studies have also determined that AMPK is also activated by mitochondrial ROS [43]. In response to ROS, AMPK leads to an upregulation of antioxidant genes, including Catalase, Sod 1, Sod2, and Ucp2. 

Since the antiestrogen 4-OHT (the active metabolite of TAM) and the antiprogesterone, mifepristone, induces mitochondrial membrane permeabilization and ROS in breast cancer cells [44], AMPK is well positioned to be a central regulator of autophagy in hormonally treated breast cancer cells [45]. In fact, a key role for TSC2/AMPK mediated mTOR inhibition was identified in early studies by the Clarke laboratory that focused on the role of the glucose-regulated protein 78 (GRP78) in autophagy induction in antiestrogen sensitive and resistant breast cancer cells [46]. In a recent study, the Koumenis laboratory identified AMPK activation as a mechanism of autophagy induction in breast cancer cells subjected to high-dose TAM [47]. In this study, the interplay of the AMPK isoforms AMPKα1 and AMPKα2 were critical in the determination of TAM-induced autophagy and cytotoxicity and AMPK α1 was identified as a requirement for TAM-induced cytotoxicity, while AMPK α2 was required for autophagy [47]. However, in an independent study, one of the changes identified in TAM-resistant breast cancer cells was a decrease in the levels of AMPK [48]. Thus, it is possible that AMPK is required for TAM-induced autophagy, but its upregulation is transient so that surviving breast cancer cells can ultimately proliferate. Further, AMPKα1 is downregulated in advanced breast cancer and is associated with poor clinical outcomes and metastasis [49]. 

These findings emphasize the need for additional studies to clarify the specific roles of the AMPK isoforms in the regulation of breast cancer autophagy, survival, and progression. If AMPK-mediated autophagy is a modality allowing breast cancer cells to survive endocrine therapy in patients, successful targeting may require a careful sequencing strategy.

## 4. The Pro-Survival Role of Autophagy in Tamoxifen Resistance

A large number of studies in the literature have investigated the relationship between the development of Tamoxifen (TAM)—based therapy resistance and autophagy. Qadir et al. [20] investigated the role of TAM-induced autophagy in different breast cancer cell lines, including ER+ (MCF-7 &T47D), and HER2 overexpressing MCF7 (MCF7-HER2) cells. Autophagy induction in response to TAM in different breast cancer cell lines was confirmed by the accumulation of GFP-LC3 puncta, monodansylcadaverine (MDC) staining as well as with the lysosomal marker, Lysotracker. Importantly, autophagy inhibition via siRNA targeting of Atg7, Atg5, and Atg8 (Beclin-1) combined with TAM caused a dramatic reduction in MCF-7 cell viability compared to that of control cells treated with non-targeting (scrambled) siRNA. Similar outcomes were reported in T47-D and MCF7-HER2 expressing cells. These results indicated that autophagy inhibition can sensitize antiestrogen-sensitive and resistant ER(+) breast cancer cells to TAM-induced cytotoxic effects, specifically mitochondrial depolarization followed by caspase-9 activation and apoptosis via the intrinsic pathway [20]. A cytoprotective role for autophagy in the actual development of antiestrogen resistance (acquired resistance) was also demonstrated utilizing MCF-7 cells and an antiestrogen resistant MCF-7 subline that was selected with a stepwise selection protocol utilizing 4-hydroxytamoxifen [21].

Following these seminal observations, Cook et al. [46], conducted in vivo studies that demonstrated a pro-survival role of autophagy in the response of antiestrogen sensitive and resistant breast cancer cells to TAM and Fulvestrant [46]. For these studies, the breast cancer cells were orthotopically injected into the mammary fat pad of female athymic mice. Mice harboring tumors of approximately 25–35 mm^2^ were treated with TAM and Fulvestrant as single agents or in combination with hydroxychloroquine (HCQ), a lysosomotropic agent that blocks the autolysosomal flux [50]. Tumor growth of the antiestrogen resistant cells, designated LCC9, was inhibited only by the combined treatment (HCQ + TAM); neither TAM nor HCQ, used as single agents, reduced the LCC9 tumor growth [51]. During the course of this study, in vitro experiments showed that HCQ potentiated the anti-tumor effect of TAM with a significant reduction in cell viability of antiestrogen sensitive and resistant cells in comparison to the cells treated with TAM as a single agent. The combined treatment of TAM and HCQ showed the characteristics of autophagy inhibition, including p62/SQSTM1 accumulation as well as increased LC3-II formation. Collectively, these results emphasize the involvement of autophagy with its cytoprotective role in TAM-resistance, setting the stage for autophagy to be considered a potential therapeutic target in ER+ breast cancer.

A link between autophagy and TAM resistance was also evident in studies where the specific upregulation of autophagy genes was identified in TAM-resistant MCF-7 breast cancer cells. For example, elevated levels of Beclin-1 (Atg6) and LC3-II (Atg8) were identified in TAM-resistant MCF-7 breast cancer cells as compared to the antiestrogen-sensitive parent cells [52]. In a similar manner, Sun et al. [53] demonstrated elevated autophagic flux in antiestrogen resistant MCF-7/TAMR1 cells compared to parental MCF-7 cells, with increases in LC3B and autophagosome number, as well as more pronounced p62/SQSTM1degradation [53]. In addition, the expression of Glucose transporter 1 (GLUT1) was elevated in the MCF-7/TAMR1 cells. Moreover, GLUT1 silencing by siRNA targeting inhibited the autophagic flux, as confirmed by reduced levels of LC3B, increased autophagosome number, and p62/SQSTM1 accumulation, accompanied by reduced the growth of 4-OHT treated MCF7/TAMR-1 cells. Of particular interest, this study identified higher expressions of both LC3B and GLUT1 in TAM-resistant clinical samples compared to levels in the TAM-sensitive counterparts. These results support the concept that autophagy plays a key role during the development of TAM resistance and highlight a *cytoprotective* role for GLUT1 in TAM-mediated autophagy [53].

Mechanistically, it is not entirely clear how autophagy contributes to the development of TAM resistance in breast cancer. It has been established that Beclin 1 upregulation reduces estrogenic signaling and thus impairs the growth response of ER+ breast cancer cells [54]. However, this regulatory role of Beclin 1 alone cannot explain the development of antiestrogen resistance. As discussed in the previous section, the AMPK pathway of autophagy may be required to prevent death due to an energy crisis or ROS. Recent studies have also focused on the role of the lysosome in the development of antiestrogen resistance. The lysosome plays a vital role in the autophagic flux which is required for the engulfed cargo to be degraded during autophagy. However, lysosomes also protect cancer cells from chemotherapeutics by promoting their intra-lysosomal sequestration and subsequent removal by exocytosis [55]. If exocytosis does not occur, the drug(s) will accumulate to levels that trigger lysosomal membrane permeabilization (LMP) and ultimately lysosomal death pathways. 

In a recent study, Actis et al. [56] showed that TAM triggered reversible lysosomal damage (LMP) in MCF-7 cells, as evidenced by the appearance of cells with faint and diffuse Lysotracker Red fluorescence rather than the bright and punctate staining of healthy lysosomes [56]. TAM treatment also triggered autophagy that was shown to protect lysosomes from LMP. Interestingly, TAM-mediated LMP in antiestrogen sensitive and resistant breast cancer cells was abrogated and viability reduced upon co-treatment with 3-methyladenine (3-MA), an early-stage autophagy inhibitor, or chloroquine (CQ), a lysosomotropic agent that blocks autolysosomal turnover. Furthermore, Actis et al. [56] identified the upregulation of a number of iron-binding proteins in antiestrogen-resistant breast cancer cells by western blotting and confocal microscopy, including ferritin heavy chain (FtH), metallothionein 2A (MT2A), and heat shock protein 70 (Hsp70). The targeting of these proteins with siRNAs significantly increased the number of cells undergoing LMP. These proteins are known to protect the lysosomal compartment against drug-induced LMP [57,58,59] and their dysregulation has been reported in some breast cancer types [60,61,62]. In the antiestrogen sensitive and resistant cells used in this study, blockade of autophagy induced LMP, further establishing a positive correlation between lysosomal integrity and functional autophagy [56]. In a similar manner, Hultsh et al. [63] found that the lysosomes of T-47D breast cancer cells which are resistant to the cytotoxic effects of TAM treatment are more abundant, larger, and more resistant to LMP induced by lysosomotropic agents than the lysosomes in the parental T-47D cells, further supporting a role for TAM-induced autophagy and lysosome stability [63]. 

## 5. Autophagy in Resistance to Selective Estrogen Receptor Degraders (SERDs) 

SERDs bind to the ER and either block estrogen from binding the receptor or alter the shape of the ER such that ER function is compromised (Figure 2). Typically, SERD binding to the ER results in ER degradation. The most common SERD is Fulvestrant, which is used occasionally as a monotherapy in early HR+ breast cancer cases in post-menopausal women. It is also used in advanced-stage breast cancer when other hormonal therapies fail. Although Fulvestrant is a well-tolerated breast cancer therapy, as is the case with other hormonal therapies, resistance is a major clinical impediment.

The role of autophagy in Fulvestrant resistance is not well detailed. There are a very limited number of studies that have investigated the relationship between Fulvestrant resistance and autophagy. Yu et. al. reported an inverse relationship between autophagy and apoptosis induction by Fulvestrant in MCF-7 cells and antiestrogen resistant MCF-7/LCC9 cells [64]. In their study, autophagy was identified by LC3-II puncta formation, LC3-II expression by western blot, and the staining of autophagic vacuoles by monodansyl cadaverine. Further, miR-214 was identified as an inhibitor of Fulvestrant-induced autophagy by reducing the expression of UPC2, a mitochondrial protein that can regulate mitochondrial ROS production [65,66]. Although UPC2- knockdown reduced autophagy and increased apoptosis, autophagy was not directly targeted in MCF-7 or LCC9 cells with small molecule inhibitors. Cook et al. [51], however, did utilize Fulvestrant as a single agent and in combination with HCQ for both in vivo and in vitro experiments conducted with the MCF-7 and LCC9 cell lines [51]. For the in vitro studies, treatment with Fulvestrant resulted in increased LC3-II with p62/SQSTM1 degradation, confirming autophagy induction. The combined treatment of HCQ and Fulvestrant resulted in a significant reduction in LCC9 and MCF-7 cell viability as compared to Fulvestrant used as a single agent. Autophagy inhibition by HCQ was demonstrated by the accumulation of LC3-II and p62/SQSTM1 in LCC9 and MCF-7 cells [51]. However, the in vivo studies performed by Cook et. al. [51] did not recapitulate the in vitro results. The combination of Fulvestrant and HCQ was less effective than HCQ treatment alone, while Fulvestrant used as a single agent showed no difference in the tumor size as compared to the controls. Cook et al. [51] provided preliminary data that cytokine production and macrophage activity may account for the differential results obtained with TAM versus Fulvestrant combined with HCQ. 

The data available with SERDs, although limited, emphasize that autophagy in endocrine resistance may be complex and involve crosstalk with the microenvironment. However, it is also possible that the targeting of autolysosomal turnover by HCQ is not an effective autophagy inhibition strategy in Fulvestrant-treated breast cancer cells. Fulvestrant-resistant breast cancer sublines, established from the MCF-7 cell line, have been shown to have high levels of autophagy with notable ATG7 upregulation and increased LC3-II/LC3-I ratio [67]. Further in vitro and in vivo studies utilizing multiple autophagy small molecule inhibitors and somatic cell genetic approaches are needed to clarify the role of autophagy in Fulvestrant resistance in breast cancer. 

## 6. Autophagy in Resistance to Aromatase Inhibitors (AIs)

The third class of hormonal therapy involves aromatase inhibitors, which block the enzyme aromatase (Figure 1). Aromatase converts androgens into estrogens via a mechanism referred to as aromatization. Blockade of aromatase is a therapy primarily used for the treatment of breast cancer in postmenopausal women that are producing small amounts of testosterone and testosterone precursors from the adrenal gland. The most commonly used aromatase inhibitors are the steroid Exemestane and the non-steroidal Letrozole and Anastrozole (Ana). Although aromatase inhibitors suppress the function of ER and reduce the risk of recurrence, therapeutic resistance is common and essentially inevitable in advanced disease [68].

Amaral et al. [69] studied autophagy as a possible strategy for overcoming Exemestane-acquired resistance using long term estrogen deprived, aromatase overexpressing ER+ MCF-7 breast cancer cells (LTEDaro ER+) which mimic late-stage acquired resistance to AIs in patients. They reported that Exemestane did not reduce the viability of LTEDaro ER+ cells, and that autophagy was induced, as confirmed by AO staining and increased LC3-II levels. Autophagy was suppressed in Exemestane-treated breast cancer cells with autophagy inhibitors that target different stages of autophagy, including Spautin-1 (SP), the pan-PI3K inhibitor Wortmannin (WT), and 3-methyladenine (3-MA). Autophagy inhibition was confirmed following treatment with each of these inhibitors by a noted decrease in AVOs generation as well as by the reduction in LC3-II turnover. The combined treatment of Exemestane plus an autophagy inhibitor resulted in a significant reduction in cell viability due to apoptosis induction, whereas apoptosis was not detected by single agent treatment with Exemestane or the respective autophagy inhibitor, indicating the possible role of autophagy in the development of Exemestane resistance in breast cancer. Moreover, the PI3K/AKT/mTOR pathway was implicated in the resistance mechanism. Exemestane treatment as a single agent did not affect the PI3K/AKT/mTOR pathway, whereas Exemestane combined with any one of the autophagy inhibitors substantially reduced PI3K expression and AKT phosphorylation. These data highlight the possibility that a cytoprotective role of Exemestane-mediated autophagy in LTEDaroER+ cells could be dependent on the PI3K/AKT/mTOR pathway in promoting Exemestane resistance [69], However; additional studies are needed to verify these results and conclusions.

Recently, Augusto et al. [70] confirmed that Exemestane induced a pro-survival autophagy in MCF-7 aromatase overexpressing cells, designated MCF-7aro. However, autophagy did not appear to play a role in the resistance developed to Ana and Letrozole. Neither Ana (10 µM, nor Letrozole (10 µM) induced autophagy, as confirmed by a lack of induction of LC3-II, no change in *SQSTM1* (p62) mRNA levels, and an unaltered number of AVOs in comparison to the control cells. Furthermore, ATG5 knockdown via siRNA in combination with Ana or Letrozole did not affect the viability of cells when compared to the cells treated with Ana or Letrozole alone. Studies utilizing the pan-PI3K inhibitor, Wortmannin (WT), at 0.1 µM, in combination with Ana or Letrozole indicated that WT did not sensitize the MCF7aro cells to Ana or Letrozole [70]. 

These data collectively indicate that Exemestane resistance mechanisms differ from those for Ana and Letrozole, at least with respect to autophagy. The contribution of cytoprotective autophagy to the survival of Exemestane treated breast cancer cells indicates that autophagy inhibition could potentially serve as a therapeutic strategy for sensitization of breast cancer to Exemestane (but not Anastrozole or letrozole). 

## 7. Autophagy in Resistance to Adjunctive Therapies Involving CDK 4/6 Inhibitors

Cell cycle checkpoints including cyclin-dependent kinases CDK4 and CDK6 are often deregulated in tumors and are considered one of the key cancer hallmarks. Selective targeting of CDK4/6 is an effective strategy which has shown promising preclinical and clinical results in numerous solid tumors [71]. CDK 4/6 inhibitors such as Palbociclib suppress cell cycle progression by interfering with CDK-cyclin complexes, blocking G1/S cell cycle transition [72,73]. While clinical advances with CDK4/6 inhibitors show promise, there are toxicities associated with this therapy including leukopenia and reversible neutropenia [72,74], in addition to the development of resistance with further disease progression [72].

Adjunctive therapy with CDK 4/6 inhibitors, primarily Palbociclib, are often utilized in combination with hormonal and anti-estrogen first-line therapies in advanced breast cancer cases. The current standard of care for metastatic ER-positive/Her2 negative breast cancer utilizes the combination of either the estrogen receptor degrader Fulvestrant or aromatase inhibitors such as Letrozole with CDK4/6 inhibitors such as Palbociclib. The combination of Letrozole with Palbociclib as an initial therapy has extended progression free survival in advanced ER+ HER2- breast cancer from 14.5 months to 27.6 months [75]. Once the disease progressed on prior endocrine therapy, the combination of Fulvestrant with Palbociclib extended progression-free survival in breast cancer patients from 4.6 to 11.2 months [75]. While these treatments represent remarkable improvements, escape from the tumor suppressive effects of these combinations appears to be inevitable, with the consequence that the patients unfortunately succumb to this disease. Mechanisms of breast cancer cell escape from CDK4/6 inhibitors involves intrinsic and acquired resistance and the mechanisms of resistance to the various CDK4/6 inhibitors may differ [76]. Whether the resistance mechanisms differ depending on the use of CDK4/6 inhibitors as single agents or as adjuvant therapy to hormonal treatments is unclear. In a recent study, intrinsic resistance to Palbociclib was linked to a lysosomal gene signature in luminal ER+ breast cancer, suggesting that in some ER+ breast cancer cells, the lysosome sequesters CDK4/6 inhibitors, as demonstrated for TNBC cells [77]. If this is a mechanism of resistance, the ability of the lysosome to sequester CDK4/6 inhibitors may be breast cancer cell and/or context dependent.

A limited number of studies suggest that autophagy contributes to ER+ breast cancer cell survival when Palbociclib is used as a single agent, but to a much lesser extent when used in combination with hormonal treatments. Studies by Vijayaraghavan et al. [72] have demonstrated that Palbociclib induces autophagy in MCF7 and T47D breast cancer cell lines, as indicated by increased MDC staining, autophagosome generation, and increased levels of LC3B-II, Atg-7, Beclin-1, BNIP3, and p62 [72]. While Beclin-1 or Atg-5 knockdown alone showed no effect on cell viability, when combined with Palbociclib, these genetic approaches for autophagy suppression significantly increased MCF7 and T47D cell sensitivity to Palbociclib. Interestingly, the suppression of autophagy was accompanied by increased senescence. In support of these findings, Palbociclib in combination with HCQ resulted in enhanced growth inhibition as well as increased cellular senescence compared to Palbociclib alone, without inducing apoptosis [72].

In vivo studies involving mice orthotopic xenografts of MCF7 breast cancer cells treated with Palbociclib demonstrated a significant reduction of the tumor volume and up-regulation of SA-ß gal activity and senescence-associated proteins [72]. In addition to senescence induction, autophagy was also induced by Palbociclib. Elevated levels of Atg-7 and increased degradation (turnover) of LC3B-II and p62/SQSTM1 were detected, along with increased autophagosome production in tumor cells. Importantly, treatment with the combination of Palbociclib and HCQ resulted in significantly smaller tumor volumes than for Palbociclib alone. Another autophagy inhibitor, Lys05, used in vivo in combination with Palbociclib, generated a similar trend to the studies utilizing HCQ with smaller tumors and prolonged survival compared to the controls [72]. These results are consistent with a cytoprotective role for Palbociclib induced autophagy.

As Palbociclib is not typically used as a single agent in breast cancer therapy but is combined with aromatase inhibitors or estrogen antagonists such as Fulvestrant, Vijayaraghavan et al. [72] conducted in vitro studies to address the role of autophagy in response to the combination of the aromatase inhibitor, Letrozole, and Palbociclib. Palbociclib plus Letrozole induced autophagy in aromatase-expressing MCF7 cells, as evidenced by MDC staining. Treatment of the aromatase-expressing MCF7 cells with Letrozole + Palbociclib in combination with HCQ resulted in more pronounced growth inhibition and suppression of colony formation as compared to Letrozole + Palbociclib in combination [72]. However, the enhancement was relatively modest as the Letrozole + Palbociclib combination was quite effective in tumor growth suppression. 

Recent in vitro studies in our own laboratory examined autophagy induced by the combination of Fulvestrant with Palbociclib in MCF-7 cells. Similar to the findings reported with Letrozole + Palbociclib, autophagy inhibition either pharmacologically with CQ or bafilomycin or genetically with ATG-5 knockdown produced a relatively modest sensitization to the combination treatment. In addition, we observed a pronounced senescence-mediated growth arrest induced by the combination of Fulvestrant + Palbociclib that could be sustained by the sequential addition of a BRD4 inhibitor, ARV-825, extensively delaying recovery (manuscript under review). Inhibitors of bromodomain-containing protein 4 (BRD4), particularly ARV-825, have demonstrated antitumor activity in multiple preclinical models, and have recently been considered as potential senolytics [78]. These observations suggest that the utilization of BET inhibitors in sequence with Fulvestrant + Palbociclib may provide a therapeutic advantage for breast cancer patients undergoing standard of care therapy.

## 8. Conclusions

In the case of individual treatment with estrogen antagonists or aromatase inhibitors, the autophagy induced by these agents is largely cytoprotective (see Table 1), lending credence to the possibility of autophagy inhibition as a strategy for improving therapeutic outcomes. These pre-clinical studies collectively supported the initiation of multiple clinical trials in which HCQ is combined with endocrine-based therapies [51]. These trials should help clarify the value of targeting autophagy to reduce breast cancer metastatic disease, at least with regard to the value of HCQ as the autophagy inhibitor. With regard to the role of autophagy in ER positive/Her 2 negative breast cancer undergoing combination treatment of Letrozole + Palbociclib or Fulvestrant + Palbociclib, additional pre-clinical studies are needed to support implementation of clinical trials. Although autophagy inhibition did enhance the antitumor response of these drug combinations, the effects were relatively modest and unlikely to significantly improve patient response. It must, however, be emphasized that the data that was generated testing the Fulvestrant + Palbociclib combination solely in cell culture and that additional studies in tumor bearing animal models will be necessary to reach more definitive conclusions. Here it should be noted that the absence of syngeneic ER positive mouse breast tumor cell lines may limit ability to translate in vivo findings to the clinic.

## Figures and Tables

**Figure 1 cancers-14-04289-f001:**
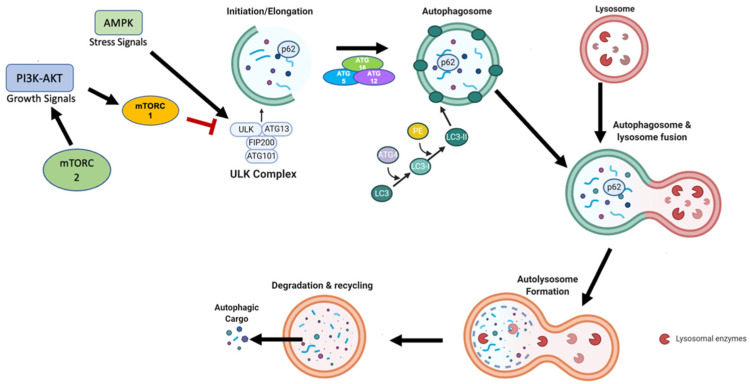
Primary mechanism of autophagy. When growth factors or nutrients become scarce, AMPK or mTOR inhibition results in activation of the ULK complex, which leads to phagophore initiation through mediation by the Beclin1 complex. The phagophore elongates and matures with the recruitment of ATG proteins, which contribute to the formation of the phosphatidylethanolamine (PE)-Conjugated LC3-II, which incorporates into the autophagosome membrane. After fusion with the lysosome, the autophagic cargo, comprising nutrients and metabolites, is degraded in the autolysosome and recycled back into the cytoplasm.

**Figure 2 cancers-14-04289-f002:**
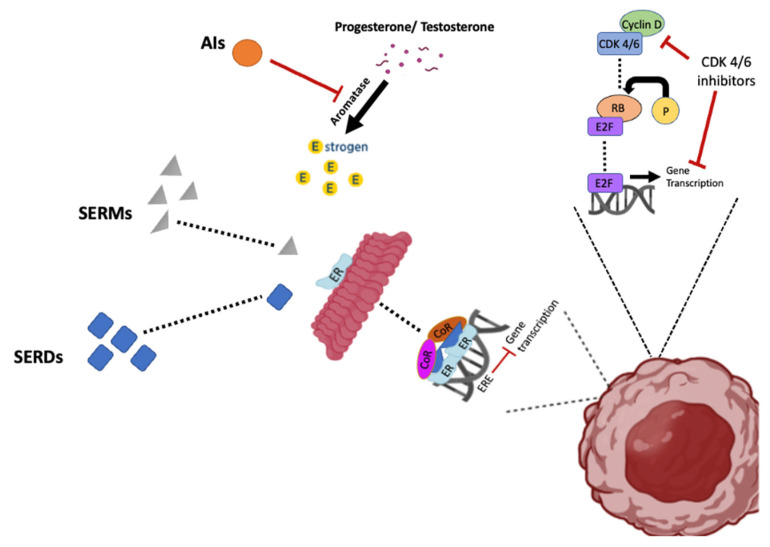
The effects of endocrine therapies (SERDs, SERMs, and AIs) and CDK 4/6 inhibitors on the ER pathway and gene transcription. SERDs block ER function by binding to the complex, resulting in degradation of the ER, SERMs prevent ER function by binding to ER to inactivate the complex, while aromatase inhibitors block ER function by inhibiting the synthesis of estradiol. CDK4/6 inhibitors prevent cell cycle progression by blocking the formation of CDK4/6 and cyclin D complex, which leads to the inhibition of gene transcription.

**Table 1 cancers-14-04289-t001:** Various functional roles of autophagy in response to endocrine therapies.

Compound	Cancer Cell Line	Nature of Autophagy	References
Tamoxifen	MCF-7, T47D ER+, and HER2 overexpressing MCF7 (MCF7-HER2) cells	Cytoprotective	[20]
	MCF-7 and antiestrogen resistant MCF-7 cells	Cytoprotective	[21]
	Antiestrogen resistant MCF-7/LCC9 cells	Cytoprotective	[52]
	Parental MCF-7 and TAM-resistant MCF-7 (TAM-R) cell lines.	Autophagy genes upregulated in the resistant cells compared to the parent cells.	[54]
	MCF-7 and antiestrogen resistant MCF-7/TAMR1 cells	Cytoprotective	[55]
	Antiestrogen sensitive and resistant MCF-7 cells	Cytoprotective	[58]
	Tamoxifen resistant and parental T-47D cells	Lysosomes are more resistant to LMP induced by tamoxifen in the resistant cells as compared to the parent cell line	[65]
Fulvestrant	MCF-7 and antiestrogen resistant MCF-7/LCC9 cells	Cytoprotective	[66]
	MCF-7 and antiestrogen resistant MCF-7/LCC9 cells	Cytoprotective in vitro Non protective in vivo	[52]
	Fulvestrant resistant MCF-7 sublines	high levels of autophagy	[68]
Exemestane	Long-term estrogen deprived, aromatase overexpressing estrogen positive MCF-7 cells (LTEDaro ER+)	Cytoprotective	[70]
	Aromatase overexpressing MCF-7aro cell line	Cytoprotective with Exemestane Non protective with letrozole and anastrozole	[71]
Palbociclib	MCF7 and T47D cell lines	Cytoprotective	[73]
Palbociclib and letrozole	Aromatase-expressing MCF7 cell line	Mild cytoprotection	[73]
Palbociclib and Fulvestrant	MCF-7 cell line	Mild cytoprotection	manuscript under review

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
