# Peer review of "Therapeutic Potential for Targeting Autophagy in ER+ Breast Cancer"

_cancers, 2022, doi:10.3390/cancers14174289_

Round 1

Reviewer 1 Report

Sample Summary stess guide-lines for providing  a concise and  understable  access  of  concepts to a large  scientific and medical  society. Even  these various writing proposals  must not be included in the present publication  ( limit  overview of the adressed topics to the Abstract), they must absolutely taken ino account  which is unfortunately not the case  for the present manuscript.  Hence,  a new formatage of the latter  is required   for  an easy access  to the enclosed important and relatively new   informations.

Paragraphs  are largely too long  without any delineations between the adressed  topics. To facilitate meaning of this statement I propose  some examples of  paragrahs cuttings  to clearly identify  the adressed informations . Line X  corresponds to a new paragraph .

#2  L71 Acquired mechanism... ; L86  Autophagy...      #3  L124 The autophagy pathway...; L147 The protein Sequestosome... ; L176 AMPK...; L183 Since the antiestrogen...      #4 L222 Following ...; L275 Futhermore, Actis... #5 L320 Collectively,...; L370 These data collectively...

 Of course other writing procecures  might be foud  to clarify the content of the manuscipt. Images  of undelrying procedures of each topic might be helpfull. Fig 1 is largelly unsufficient in this regard .  Clinical investigations data  should clearly be identified from studies performrd with cell lines. 

I realise  that ssuch a formatting  would be time consuming , but   it is  necessary to accept the conclusions  which would be explicitely related  to the data of Table 1. 

 Complementary remarks . Table 1   N/A  explicit  meaning;  Ref 40 typography

Author Response

  1. Sample Summarystess guide-lines for providing  a concise and  understable  access  of  concepts to a large  scientific and medical  society. Even  these various writing proposals  must not be included in the present publication  ( limit  overview of the adressed topics to the Abstract), they must absolutely taken ino account  which is unfortunately not the case  for the present manuscript.  Hence,  a new formatage of the latter  is required   for  an easy access  to the enclosed important and relatively new   informations.

A: Thank you for your guidance; we limited the overview of our topic to the Abstract.

  1. Paragraphs are largely too long without any delineations between the addressed topics. To facilitate meaning of this statement I propose some examples of paragraphs cuttings to clearly identify the addressed information. Line X corresponds to a new paragraph.

#2  L71 Acquired mechanism... ; L86  Autophagy...      #3  L124 The autophagy pathway...; L147 The protein Sequestosome... ; L176 AMPK...; L183 Since the antiestrogen...      #4 L222 Following ...; L275 Futhermore, Actis... #5 L320 Collectively,...; L370 These data collectively...

A: Thank you for the recommendation. We corrected the length of all the paragraphs in the manuscript as per your recommendations.

  1. Of course other writing procedures might be found to clarify the content of the manuscript. Images of underlying procedures of each topic might be helpfull. Fig 1 is largelly unsufficient in this regard.  Clinical investigations data should clearly be identified from studies performed with cell lines. 

A: Thank you for your guidance; we corrected it, but the goal of it is to give the readers idea about the mechanism of actions of the discussed drugs.

  1. I realize that such a formatting would be time consuming, but   it is necessary to accept the conclusions which would be explicitly related to the data of Table 1. Complementary remarks. Table 1   N/A  explicit  meaning;  Ref 40 typography

A: Thank you for your guidance; we updated the table.

Reviewer 2 Report

Article entitled "Therapeutic Potential for Targeting Autophagy in ER+ Breast Cancer" described the role of autophagy in the therapeutic intervention of ER+ Breast Cancer but the authors fails to nicely correlate the detailed molecular mechanism that leads to clearance/inhibition of ER+ Breast Cancer.

Few points needs to add before further decision about the article:

1. Add detailed about the recruitment of autophagy protein LC-3 II to phagosome and maturation of phagosome. Read following paper related to autophagy molecular mechanism and modify article accordingly by citing few more relevant papers

https://pubmed.ncbi.nlm.nih.gov/27814595/

2. Triple-negative breast cancer is more aggressive than ER-positive which is missing in the article. Please revise by adding it

3. Figure 1 seems to be copied please redraw it by incorporating detailed mechanistic events involved inhibition of cancer progression.

4. Based upon the content mentioned on page no.4 Lines no. 127 to 180 develop one table or graphics for better understanding by readers.

5. Correlate the in vivo mouse models such as the xenograft model developed through injecting MDA-MB 231 and how autophagy is leading to inhibition of the tumor progression.

6. In the revised version authors must have to focus on triple-negative breast cancer, autophagy, and inhibition of cell cycle cycle/signaling pathways.

7. There is several inconsistent spacing throughout the manuscript so please carefully revised it

Author Response

  1. Add detailed about the recruitment of autophagy protein LC-3 II to phagosome and maturation of phagosome. Read following paper related to autophagy molecular mechanism and modify article accordingly by citing few more relevant papers

A: We included more details regarding this part and cited relevant papers.

  1. Triple-negative breast cancer is more aggressive than ER-positive which is missing in the article. Please revise by adding it

A: Thank you for your suggestion, but our focus in this manuscript is directed to ER+ breast cancer, as indicated in the title of the paper.

  1. Figure 1 seems to be copied please redraw it by incorporating detailed mechanistic events involved inhibition of cancer progression.

A: Thank you for your comment; we corrected it, but the goal of this figure is to give the readers idea about the mechanism of actions of the discussed drugs.

  1. Based upon the content mentioned on page no.4 Lines no. 127 to 180 develop one table or graphics for better understanding by readers.

A: Thank you for your suggestion, but the molecular mechanism of autophagy is already graphed by many papers in the literature, and we are cited quite a few papers which can be referenced for the readers to go through it. We can add one if it is necessary.

  1. Correlate the in vivo mouse models such as the xenograft model developed through injecting MDA-MB 231 and how autophagy is leading to inhibition of the tumor progression.

A: Thank you for your suggestion, but our focus in this manuscript is directed to ER+ breast cancer as indicated in the title.

  1. In the revised version authors must have to focus on triple-negative breast cancer, autophagy, and inhibition of cell cycle cycle/signaling pathways.

A: Thank you for your suggestion, but again, our focus in this manuscript is directed to ER+ breast cancer and the role of autophagy mediated by different endocrine and adjuvant therapy, used clinically for the treatment of ER+ breast cancer patients, and the therapeutic potential for targeting autophagy to increase the effectiveness of ER+ breast cancer therapies.

  1. There is several inconsistent spacing throughout the manuscript so please carefully revised it

A: Thank you for your guidance; we corrected the spacing between paragraphs.

Reviewer 3 Report

The authors provide a review of literature focusing on the therapeutic potential for targeting autophagy in ER+ breast cancer.

"Autophagy in resistance to Ais" can be a separate section, i.e. section 6 then the "Autophagy in Resistance to Adjunctive Therapies involving cdk4/6 inhibitors" section will be section 7

Author Response

  1. "Autophagy in resistance to Ais" can be a separate section, i.e. section 6 then the "Autophagy in Resistance to Adjunctive Therapies involving cdk4/6 inhibitors" section will be section 7

A: Thank you for your guidance; we separated them into separate sections.

Reviewer 4 Report

In this manuscript, the autors a review of the resistance mechanisms involved in endocrine terapies.
The authors discuss the current data situation in detail, nevertheless, the following comments should be considered before publication.
1. The third chapter describes in detail the mechanism of autophagy, hence I propose to change its title from "Mechanisms of Macroautophagy (Autophagy)" to "Mechanisms of Autophagy".
2. Line 243, 382, 392 and 395: incorrect reference to literature also changed the citation format
3. In my opinion the caption of table 1 is incorrect.

Author Response

  1. The third chapter describes in detail the mechanism of autophagy, hence I propose to change its title from "Mechanisms of Macroautophagy (Autophagy)" to "Mechanisms of Autophagy".

A: Thank you for your suggestion. We changed the title from Mechanisms of Macroautophagy (Autophagy)" to "Mechanisms of Autophagy".

  1. Line 243, 382, 392 and 395: incorrect reference to literature also changed the citation format.

A: Thank you for your guidance; we have corrected the references and references format.

  1. In my opinion the caption of table 1 is incorrect.

A: Thank you for your guidance; we have corrected the table caption.

Round 2

Reviewer 1 Report

 Authors   accept my request  for  a  strong  delineation of the  various adressed topics  to clearly state  their implication  in autophagy.   Access to these topics is now largely  more easy which justifies my acceptance  of the manuscript for publication. Nevertheless, I slill  consider that an inclusion  of a complementary iconograhy related to the various  mechanismss evoked in section 3  ( one for each paragraph  starting to the second one) would  facilitate  access to these  mechanisms .  An underlying footnote might be helpfull in this regard ( for  details see text ). 

Complementary remark of minor importance . Since a reference to Fig1 i is icluded within the text  for CERD (§5) and AIs (§6)  do the same for Tamoxifen (§4) and CDK4/6(§7) . 

Author Response

Thank you again for your guidance. I have included a basic figure and legend of the autophagy mechanism.

Reviewer 2 Report

The authors have addressed all the issues and nicely revised the manuscript. In my opinion now can be consider for the publication

Author Response

Thank you.